# ConvexBench: Can LLMs Recognize Convex Functions?

Yepeng Liu [1]   Yu Huang [2]   Yu-Xiang Wang [3]   Yingbin Liang [4]   Yuheng Bu [1]

## Abstract

Convex analysis is a modern branch of mathematics with many applications. As Large Language Models (LLMs) start to automate research-level math and sciences, it is important for LLMs to demonstrate the ability to understand and reason with convexity. We introduce ConvexBench, a scalable and mechanically verifiable benchmark for testing *whether LLMs can identify the convexity of a symbolic objective under deep functional composition*. Experiments on frontier LLMs reveal a sharp compositional reasoning gap: performance degrades rapidly with increasing depth, dropping from an F1-score of 1.0 at depth 2 to approximately 0.2 at depth 100. Inspection of models' reasoning traces indicates two failure modes: *parsing failure* and *lazy reasoning*. To address these limitations, we propose an agentic divide-and-conquer framework that (i) offloads parsing to an external tool to construct an abstract syntax tree (AST) and (ii) enforces recursive reasoning over each intermediate sub-expression with focused context. This framework reliably mitigates deep-composition failures, achieving substantial performance improvement at large depths (e.g., F1-Score = 1.0 at depth 100). Our code is available at https://github.com/yepengliu/ConvexBench.

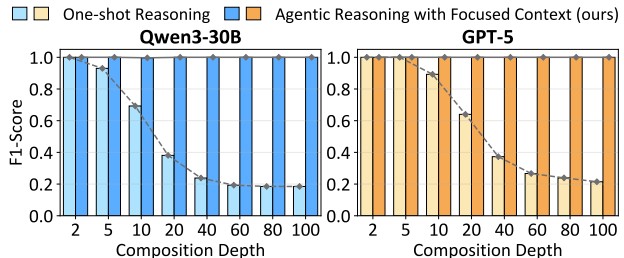

*Figure 1.* F1-Score on ConvexBench versus composition depth for Qwen3-30B and GPT-5, comparing one-shot reasoning to our agentic reasoning with focused context. One-shot reasoning performance drops from 1.0 at shallow depth to around 0.2 at depth 100, while the agentic framework maintains 1.0 across depths.

## 1. Introduction

The emerging paradigm of AI for Research (Novikov et al., 2025; Wei et al., 2025) envisions Large Language Models (LLMs) as capable assistants that can automate complex mathematical reasoning (Georgiev et al., 2025; Yang et al., 2024b) and scientific workflows (Chen et al., 2025b). A key capability in this paradigm is the ability of LLMs to accurately understand and analyze complex symbolic expressions (Mirzadeh et al., 2024), such as identifying the convexity of a function in optimization problems.

Many symbolic objectives encountered in practice are not given as single explicit expressions, but are constructed incrementally through multiple modeling steps (Diamond & Boyd, 2016). For example, starting from simple terms, objective functions are often built by applying smoothing to non-smooth components (Nesterov, 2005), introducing penalty or barrier formulations to handle constraints (Boyd & Vandenberghe, 2004), and wrapping intermediate expressions to improve numerical stability or robustness. Repeating such transformations, often through modular reuse of previously defined sub-expressions, naturally produces objectives with deeply compositional structure.

Reasoning about the convexity of a deeply composed function requires verifying domain, monotonicity, and convexity conditions at *every level* of composition. This can be very tedious for humans, and a single local mistake invalidates all downstream conclusions (Dziri et al., 2023). Recent LLMs have exhibited strong performance on mathematical reasoning (Shao et al., 2024; Luo et al., 2023; Yang et al., 2024a), making them seemingly well-suited to this analysis, which involves the application of simple rules. However,

*Can LLMs recognize convex functions*
*as composition depth increases?*

With this motivation, we introduce ConvexBench, a scalable and mechanically verifiable benchmark of compositional functions with controlled depth. Following the Disciplined

---

[1]UC Santa Barbara [2]University of Pennsylvania [3]UC San Diego [4]The Ohio State University. Correspondence to: Yepeng Liu <yepengliu@ucsb.edu>.

*Proceedings of the 43rd International Conference on Machine Learning*, Seoul, South Korea. PMLR 306, 2026. Copyright 2026 by the author(s).

Convex Programming (DCP) paradigm (Grant et al., 2006), we generate objectives by composing convex, concave, and affine atoms under certified composition rules. This design offers two key advantages: (i) it yields mechanically verifiable labels - for any generated expression, we can automatically determine its convexity using a rule-based checker, avoiding noisy human annotation; and (ii) by repeatedly composing atoms and rules, we can control the depth and the size of the expression tree, thereby creating a smooth axis of compositional difficulty while keeping each local reasoning step elementary.

Our experiments on ConvexBench reveal a *compositional reasoning gap* in current LLMs. Models achieve perfect performance on shallow expressions (e.g., F1-Score = 1.0, when Depth = 2), but performance degrades rapidly as compositional depth increases, beginning as early as depth 5 and dropping to approximately 0.2 at depth 100. However, the problem does not require advanced convex analysis: each instance can be solved by repeatedly applying a small set of standard DCP rules. An analysis of model outputs suggests two recurring failure modes:

1. **Parsing failures:** models frequently lose track of parentheses and operator scope, misidentify sub-expressions, or conflate independent terms, which leads to incorrect applications of composition rules.

2. **Lazy reasoning:** models tend to rely on shallow heuristics rather than step-by-step reasoning, e.g., by assuming unverified properties, focusing on only a small sub-expression while ignoring the rest, or abandoning the analysis due to structural complexity.

To address these bottlenecks, we propose an **agentic divide-and-conquer framework** that enforces step-by-step reasoning for deeply composed objectives. Specifically, we first introduce a *tool-integrated decomposition*, which offloads structural parsing of complex expressions to an external tool, and deterministically parses the function into an abstract syntax tree (AST). Providing LLMs directly with an AST (see Figure 3 (2)) leads to a significant performance improvement (Figure 4a and 4b), particularly for more advanced models. However, even with perfect parsing, the one-shot reasoning cannot ensure step-by-step verification for each component: a single unchecked inference can propagate through the reasoning chain and ultimately lead to an incorrect conclusion. Therefore, we design an *agentic reasoning* system (see Figure 3 (3) and (4)) that analyzes the expression recursively with *focused context*, explicitly verifying intermediate states before composing results. In extensive experiments on ConvexBench (Figure 1), one-shot reasoning collapses at depth 100 (F1-Score $\approx 0.2$ for Qwen3-30B and GPT-5), whereas our agentic framework achieves F1-Score = 1.0 for both models.

We summarize our contributions and key findings below:

1. We develop ConvexBench, a benchmark of convex and nonconvex functions constructed from DCP-style atoms and certified composition rules. All instances come with mechanically verified labels and controlled complexity composition depth.

2. We benchmark a range of frontier LLMs on ConvexBench and identify two failure modes: (i) brittle structural parsing of long expressions, and (ii) lazy reasoning that fails to propagate composition rules through the full expression tree.

3. We propose three agentic frameworks (Figure 3 (2), (3), and (4)) to address the bottlenecks. Specifically, the agentic reasoning with focused context approach substantially improves performance on deeply compositional instances, closing the gap observed under one-shot reasoning.

## 2. Related Works

**Long-context LLMs.** Long-context LLMs have become increasingly important, as many applications require models to retain, retrieve, and reason over information spread across lengthy inputs (Liu et al., 2025; Wan et al., 2025b). A major bottleneck in this setting is the poor scalability of standard self-attention: both computation and memory usage grow rapidly with the context length, which motivates a rich literature on more efficient attention mechanisms (Dao et al., 2022; Dao, 2023), architectural modifications (Sun et al., 2024; Gu & Dao, 2024; Peng et al., 2023), and adjustments to position embeddings (Xiong et al., 2024; Zhu et al., 2023) for long sequences. In parallel, another line of work investigates long-horizon reasoning under long contexts, asking how model performance changes as problems demand more steps, deeper composition, and longer dependency chains (Shojaee et al., 2025; Meyerson et al., 2025; Chen et al., 2025a; Malek et al., 2025). This has also spurred a variety of benchmarks targeting long-context understanding and long-range reasoning (Hsieh et al., 2024; Bai et al., 2025; Kuratov et al., 2024; Loughridge et al., 2024; Zhou et al., 2025; Ling et al., 2025; Mirzadeh et al., 2024). Our work contributes to this evaluation-focused direction by introducing a benchmark that probes long-horizon compositional capabilities, specialized to convex optimization tasks.

**Multi-Agent LLMs for Long-Horizon Tasks.** Multi-agent LLM systems orchestrate multiple interacting agents with specialized roles (e.g., planning, execution, and verification) to solve complex tasks through coordination and iterative dialogue (Wu et al., 2024; Guo et al., 2024). Such systems frequently operate over long horizons, where the interaction history grows over time and effective context management becomes a central challenge (Yao et al., 2022; Park et al., 2023). Existing strategies are often organized around two themes: (i) summarizing or distilling past trajectories to

stay within a fixed context budget (Tang et al., 2025; Wang et al., 2023; Yu et al., 2026), and (ii) leveraging collaboration among agents to share responsibility for memory and decision-making (Anthropic, 2025; Zhang et al., 2024; Wong et al., 2025; Wan et al., 2025a). Our work aligns with this line of research by embedding adaptive context selection directly into the reasoning process, so that the system dynamically focuses on the most task-relevant information to maintain adaptivity over long horizons.

**LLMs for mathematical reasoning.** LLMs have recently demonstrated substantial progress in mathematical reasoning (Wan et al., 2026; Shao et al., 2024; Zhao et al., 2026b; Kang et al., 2026; Zhang et al., 2026), spanning informal natural-language problem solving (e.g., OpenAI o3 (OpenAI, 2024), DeepSeek-R1 (Guo et al., 2025)) and formal theorem proving (e.g., AlphaProof (Hubert et al., 2025), BFS-Prover (Xin et al., 2025)). This rapid progress motivates increasingly discriminative evaluations (Lu et al., 2025; Sun et al., 2025; Zhao et al., 2026a). Classic short-answer benchmarks such as MathQA (Amini et al., 2019), GSM8K (Cobbe et al., 2021), and MATH (Hendrycks et al., 2021) are approaching saturation, while harder competition-style problems (e.g., Omni-MATH (Gao et al., 2024)) also see fast gains. However, high final-answer accuracy can obscure brittleness in intermediate reasoning, especially for problems requiring many dependent steps. Recent benchmarks, therefore, emphasize long-horizon and compositional generalization, reporting a persistent *reasoning gap* where reliability degrades with the length of the reasoning chain or functional nesting (Mirzadeh et al., 2024; Wang et al., 2024; Zhou et al., 2025; Sinha et al., 2025). Our work extends this direction to convex optimization by introducing ConvexBench, which stress-tests LLMs' ability to analyze complex symbolic expressions.

## 3. ConvexBench

The primary goal of ConvexBench is to evaluate the symbolic expression reasoning capabilities of LLMs, through the lens of the convexity identification problem of compositional functions with varying depths. To achieve this goal, the constructed benchmark should satisfy the following properties: (1) **Verifiable**: ConvexBench ensures ground-truth reliability by employing the DCP-compliant synthesis procedure, supplemented by numerical validation via Jensen's test. (2) **Scalable**: ConvexBench provides a controllable synthesis pipeline, allowing 1 modulated by the compositional depth.

### 3.1. Overview and Background

The core task of ConvexBench is convexity identification: given a symbolic expression of a function $f : \mathcal{D} \to \mathbb{R}$, an LLM must determine whether $f$ is convex, concave,

or neither (neither convex nor concave) over the specified domain $\mathcal{D}$. Formally, a function $f$ is convex if for all $x$, $y \in \mathcal{D}$ and $\lambda \in [0, 1]$, the following inequality holds:

$$f(\lambda x + (1 - \lambda)y) \le \lambda f(x) + (1 - \lambda)f(y), \quad (1)$$

and concave if the inequality is reversed. To synthesize the compositional function, we first define an atom library $\mathcal{A} = \{a_1, a_2, ..., a_n\}$ consisting of a diverse set of elementary functions (e.g., exponential, logarithmic, affine, etc). To ensure the mathematical rigor of the composition, each atom $a$ is characterized by property $\tau = (\phi, \gamma, \mu, \mathcal{R})$, where: $\phi : \mathcal{D} \to \mathbb{R}$ is the symbolic mapping defining the function, $\gamma \in \{$convex, concave, affine, neither$\}$ denotes the convexity of $a$ on its domain $\mathcal{D}$, $\mu \in \{$increase, decrease, non-monotonic$\}$ denotes the monotonicity of $a$ on $\mathcal{D}$, $\mathcal{R}$ denotes the range of the function, such that $\mathcal{R} = \{\phi(x) \mid x \in \mathcal{D}\}$.

A compositional function $F^{(D)}$ of depth $D$ is defined recursively as:

$$F^{(d)}(x) = f_d(F^{(d-1)}(x)), \quad \text{for } d = 1, \dots, D, \quad (2)$$

where $F^{(0)}(x) = x$ is the identity function, and $f_d \in \mathcal{A}$ is the atom chosen at depth $d$. The complexity of verifying the convexity of $F^{(D)}$ scales with the depth $D$.

While LLMs can easily recognize convexity in shallow expressions (e.g., $D = 2$), extending this capability to larger depths (e.g., $D \ge 5$) requires step-by-step reasoning under DCP rules. It involves a sequence of local checks: identifying the convexity and monotonicity of *each* constituent atom and propagating these properties through the nested composition structure. Errors at any intermediate step can propagate and lead to an incorrect conclusion. Consequently, accurate convexity identification at large depths depends on reliable single-step reasoning throughout the expression, raising the question of whether current LLMs can consistently carry out such recursive reasoning.

### 3.2. Dataset Synthesis

In this section, we detail the procedure for generating $F^{(D)}$, as shown in Figure 2. The synthesis process follows DCP rules, ensuring that every synthesized function has a verifiable ground truth label.

To generate an expression of depth $D$ with target convexity label $\Gamma \in \{$convex, concave, neither$\}$, we sample compositions recursively under DCP constraints. We start from a base atom $f_1 \in \mathcal{A}$ and iteratively wrap it with outer atoms: $F^{(1)} = f_1$ and $F^{(d)} = f_d(F^{(d-1)})$ for $d = 2, ..., D$. At each layer $d$, we maintain a state $S^{(d)} = \{\phi_{F^{(d)}}, \gamma_{F^{(d)}}, \mu_{F^{(d)}}, \mathcal{R}_{F^{(d)}}\}$ (e.g., expression, convexity, monotonicity, and range). The next outer atom $f_d$ is chosen by DCP-guided sampling rule $\pi$ conditioned on the previous state and targets: $f_d \leftarrow \pi(S^{(d-1)}, \Gamma, D)$.

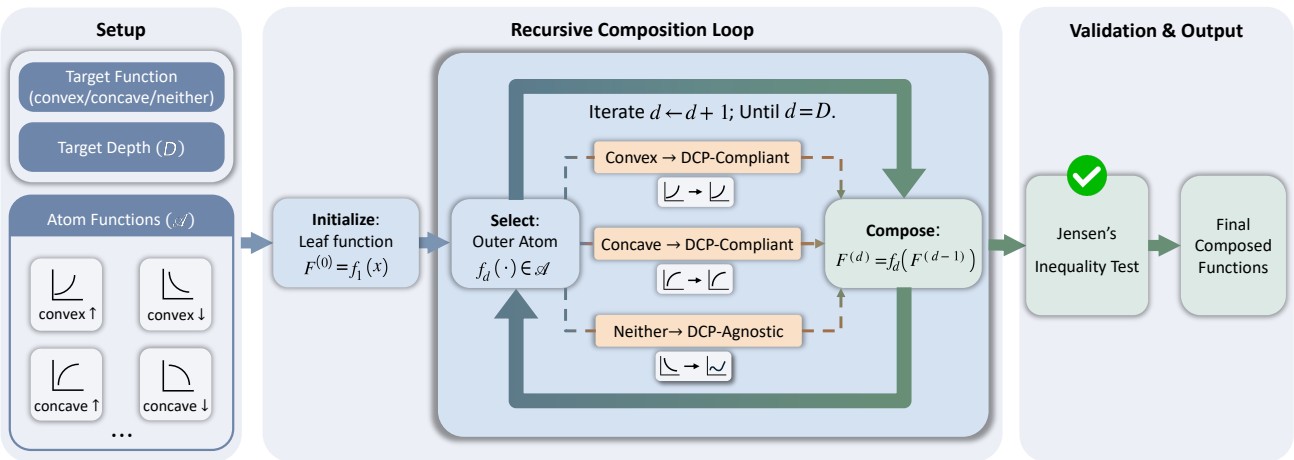

Figure 2. Overview of ConvexBench construction. We recursively compose atoms from $\mathcal{A}$ to reach the target depth and convexity, producing an expression with controlled composition depth. For convex/concave targets, outer atoms are chosen to satisfy DCP rules; for neither class, we relax DCP constraints and admit a constructed function only after Jensen's Inequality test finds counterexamples.

For $\Gamma \in$ {convex, concave}, $\pi$ conducts **DCP-compliant sampling**. Specifically, for each intermediate layer $d < D$, it samples an outer atom $f_d \in \mathcal{A}$ that is compatible with the current state (i.e., satisfies the certified composition rules given $S^{(d-1)}$). To increase diversity, we do not constrain the intermediate convexity to match the final target: $\gamma_{F^{(d)}}$ can be convex or concave as long as each composition step is DCP-valid. At the final layer $d = D$, $\pi$ restricts to atoms $f_D$ such that composing $f_D$ with $F^{(D-1)}$ yields the target label. For $\Gamma \in$ {neither}, at each layer $d \leq D$, $\pi$ samples $f_d$ from $\mathcal{A}$ without enforcing DCP constraints, and the resulting expression is assigned the 'neither' label only if Jensen's inequality counterexample can be found.

Finally, we numerically validate all synthesized functions with a Jensen's-inequality test (see (1)). For $\Gamma \in$ {convex, concave}, convexity/concavity is guaranteed by construction via certified DCP rules, and Jensen's test is a post-hoc sanity check. For $\Gamma \in$ {neither}, we use Jensen's test as a counterexample filter, admitting a function only if it violates both convexity and concavity.

## 4. Evaluation and Method

### 4.1. Evaluation of One-shot Reasoning

With ConvexBench, we establish a one-shot reasoning baseline (Figure 3 (1)) in which an LLM receives the raw symbolic expression as input and determines its convexity: $y \leftarrow \mathcal{M}(F^{(D)})$. Ideally, in this baseline, LLMs should decompose the complex expression and apply DCP rules step-by-step within a single pass. However, as shown in Figure 4a, performance degrades sharply as composition depth increases, with the drop beginning as early as $D = 5$. We examine the reasoning processes and identify two primary causes of this performance degradation.

**Parsing Failure:** The first failure mode is primarily syntactic. As the symbolic expression grows in length and nesting depth, models frequently lose track of parentheses and operator scope, producing an incorrect decomposition of the expression (e.g., treating $g(h(x)+k(x))$ as $g(h(x))+k(x)$). Such mis-parsing corrupts downstream reasoning: even with correct DCP knowledge, applying composition rules to an incorrect expression can lead to a wrong conclusion (see example in Table 3 in Appendix).

**Lazy Reasoning:** Figure 4e shows a non-monotonic trend in the number of reasoning tokens as composition depth increases. Token counts rise initially at small depths, consistent with the model attempting to track the compositional structure, but beyond a depth threshold, they plateau or drop sharply. Qualitative inspection over the reasoning process (see example in Table 4 in Appendix) suggests that this change coincides with a shift in strategy: instead of recursively verifying intermediate steps along the expression, the model increasingly relies on local cues to guess the global properties, producing unsupported conclusions.

### 4.2. Agentic Divide-and-Conquer Frameworks

These two failure modes motivate our agentic frameworks. (1) To mitigate parsing failure, we introduce a tool-integrated decomposition stage that parses long expressions into an explicit AST (Figure 3 (2)). (2) To mitigate lazy reasoning, we introduce an agentic reasoning strategy that performs recursive, step-by-step reasoning over each sub-expression (Figure 3 (3) and (4)).

**One-shot Reasoning with Decomp.** To isolate reasoning errors from parsing failures, we provide the LLM with an explicit decomposition of the symbolic expression, rather than requiring it to parse a complex function internally. We

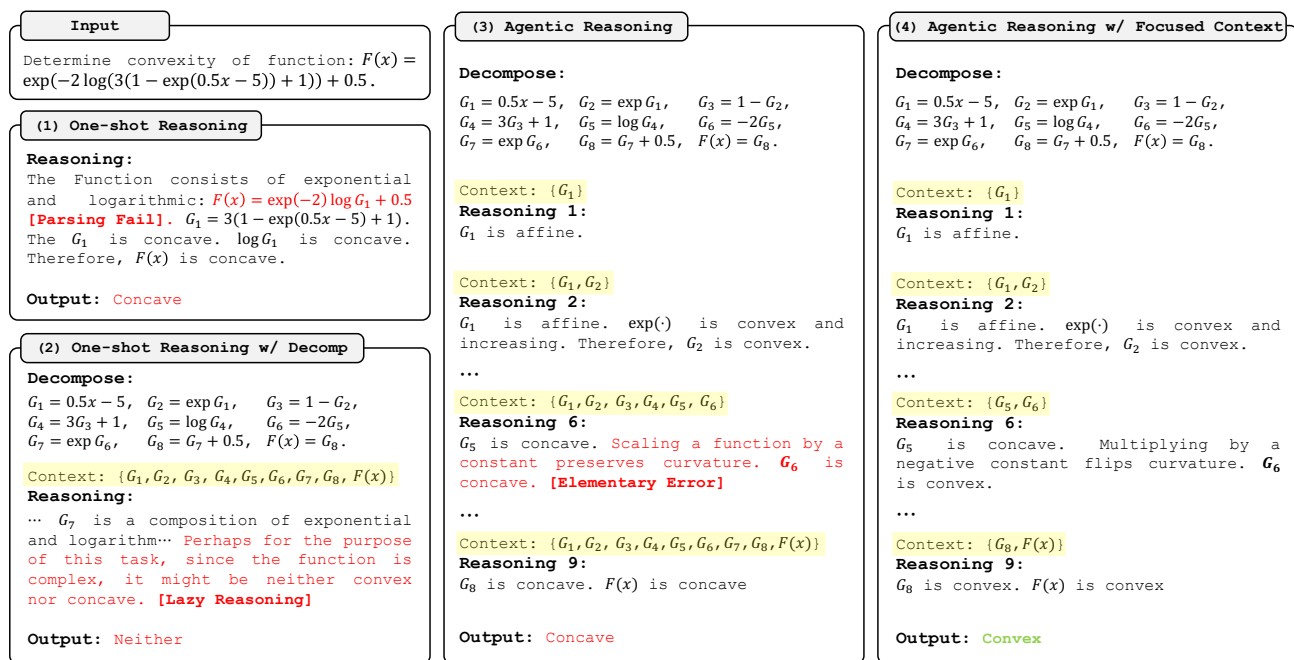

*Figure 3.* Comparison of different reasoning paradigms on ConvexBench. (1) One-shot Reasoning (Baseline) directly inputs the raw expression into LLMs. (2) One-shot Reasoning with Decomp first decomposes the raw expression into AST, then inputs the AST into LLMs. (3) Agentic Reasoning decomposes the expression into a sequence of sub-tasks and conducts recursive reasoning over each sub-task with full context. (4) Agentic Reasoning with Focused Context constructs a dependency-focused context for each sub-task.

offload the parsing process to an external tool that converts a function $F^{(D)}$ into a sequence of intermediate sub-functions $G = \{g_1, g_2, \ldots, g_k\}$, where each $g_i$ depends only on the input $x$ (e.g., $g_1 = 0.6x + 0.8$) and previously defined sub-expressions (e.g., $g_3 = g_1 + g_2$). This explicit decomposition reduces parenthesis/scope errors and isolates downstream reasoning from parsing failures in the one-shot reasoning baseline.

In this paradigm, instead of providing the raw expression $F^{(D)}$, we feed the model the entire decomposed sub-functions $C_G = \{g_i \mid 1 \leq i \leq k\}$, and ask it to determine the final convexity: $y \leftarrow \mathcal{M}(C_G)$.

As shown in Figure 4b, this structured input substantially improves performance over the one-shot reasoning baseline, especially for stronger models, consistent with parsing errors being a key failure mode. However, even with this structure, one-shot performance still degrades as composition depth increases, particularly for less capable models such as Qwen3-8B. Through a careful inspection of the reasoning traces, we make the following observations:

1. Accurate decomposition encourages more explicit intermediate-step reasoning over sub-expressions, even at larger steps.

2. Despite this improvement, LLMs remain susceptible to lazy reasoning, particularly smaller models such as Qwen3-8B, where reasoning token usage plateaus be-

yond a depth threshold (Figure 4f), indicating reduced effort as complexity increases.

3. Even when explicitly checking intermediate states, LLMs frequently make elementary errors, causing the reasoning process to collapse at intermediate stages.

**Agentic Reasoning.** Although explicit decomposition removes parsing errors, one-shot reasoning still often fails at larger depths due to incomplete intermediate verification and elementary mistakes. This motivates an agentic framework that performs recursive, step-by-step checks over the decomposed sub-expressions. Rather than asking the model to reason over the entire decomposed sequence $C_G$ in a single pass, we split a depth-$D$ composition into $k$ local tasks and require an explicit check of each intermediate sub-expression $g_i$ before composing results upward. This *recursive reasoning* prevents the model from reaching a conclusion without traversing the full expression, mitigating the lazy-reasoning failure observed in one-shot reasoning.

Specifically, for each sub-function $g_i$, an LLM performs localized reasoning and returns a state $\sigma_i \leftarrow \mathcal{M}(C_G^{(i)}, g_i)$, where $\sigma_i$ includes properties (e.g., convexity and range) of $g_i$. The context $C_G^{(i)}$ includes all previous expressions and states $C_G^{(i)} = \{(g_j, \sigma_j) \mid 1 \leq j < i\}$.

As in Figure 4c, Agentic Reasoning further improves performance over one-shot reasoning (with composition). While Agentic Reasoning enforces the intended reasoning trajec-

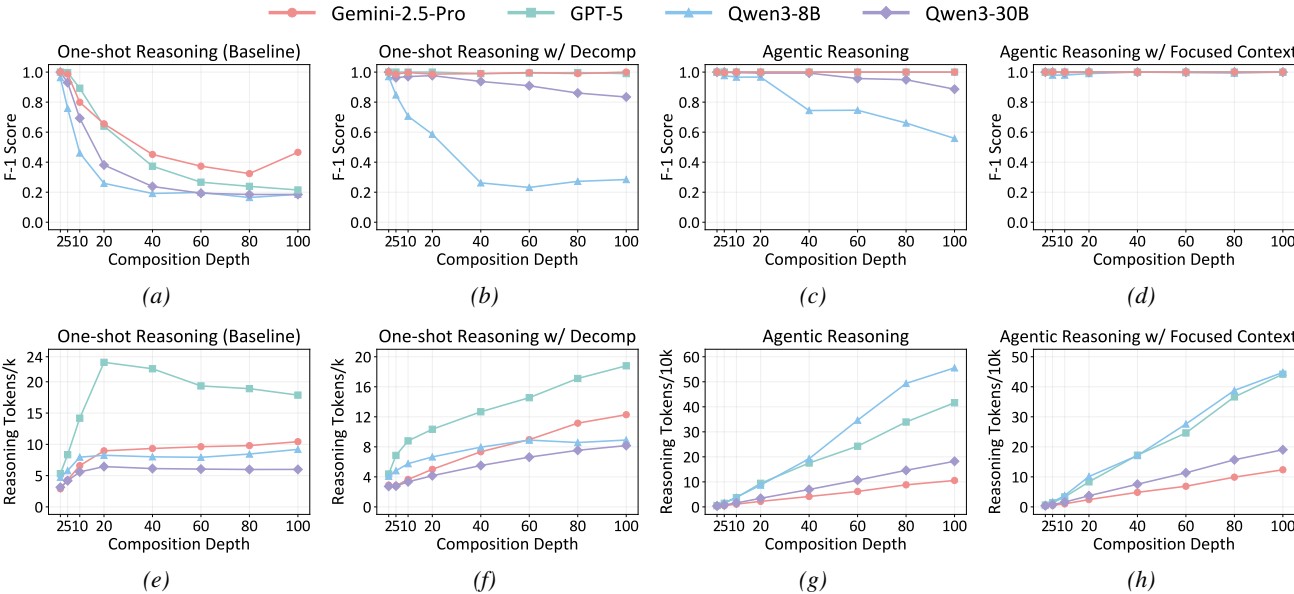

*Figure 4.* Evaluation of reasoning performance and consumed tokens across different compositional depths and models. The top row (a-d) shows the F1-Score under different reasoning paradigms. The bottom row (e-h) illustrates the average reasoning tokens consumed.

tory, it introduces a new challenge: attention distraction in long contexts. As the context $C_G^{(i)}$ grows, it accumulates redundant information that is unnecessary for the current decision (Wu et al., 2025; Shi et al., 2023). Although the context is still relatively short (mostly under 5,000 tokens as shown in Table 1), this redundancy lowers the signal-to-noise ratio, introducing distractors that can cause attention drift and cumulative errors in long-horizon chains.

**Agentic Reasoning with Focused Context.** To address the attention distraction and information redundancy inherent in the cumulative context of Agentic Reasoning, we propose Agentic Reasoning with Focused Context, which leverages the structure of $G$ to construct a dependency-focused context for each sub-task. At each step, the sub-function $g_i$ depends only on a small set of previously defined sub-functions, rather than the entire history. We thus construct the dependency-focused context: $\bar{C}_G^{(i)} = \{(g_j, \sigma_j) \mid j \in \mathrm{Pa}(i)\}$, where $\mathrm{Pa}(i)$ denotes the direct dependencies of $g_i$ in the expressions tree, and $\sigma_i \leftarrow \mathcal{M}(\bar{C}_G^{(i)}, g_i)$.

# 5. Experiments

## 5.1. Experiment Setting

**Dataset and Models.** Our experiments involve four frontier LLMs, consisting of both proprietary and open-source models: gpt-5 (Singh et al., 2025), gemini-2.5-pro (Comanici et al., 2025), qwen3-8b (Guo et al., 2025), and qwen3-30b (Yang et al., 2025). For ConvexBench, we synthesize three classes of functions: convex, concave, and

neither, each consisting of 100 samples. For each function class, we evaluate composition depths of 2, 5, 10, 20, 40, 60, 80, and 100. The atoms we use to construct ConvexBench are listed in Table 5.

**Implementation Details.** The sampling temperature is set to 0.1 for all LLMs, except for gpt-5, for which the temperature is not adjustable. For gpt-5, we use the high reasoning efforts. For open-source LLMs, the final decision is determined using a self-consistency strategy with majority voting over $N = 64$ sampled responses, whereas proprietary LLMs rely on a single sample. The maximum reasoning token number is set to $50,000$. The prompts we use are provided in Appendix B.

**Evaluation Metrics.** We consider a three-class classification task with labels convex, concave, and neither. We use the macro-averaged F1-score to evaluate the overall performance. In addition, we report one-vs-rest recall for each class to provide a fine-grained analysis of model behavior.

## 5.2. Results and Analysis

### 5.2.1. MAIN RESULTS

**Agentic Reasoning with Focused Context closes the compositional reasoning gap.** The performance of Agentic Reasoning with Focused Context on ConvexBench is shown in Figure 4d. Compared with One-shot Reasoning (Figure 4a), our method yields large gains across different depths, improving F1-score by 0.79, 0.54, and 0.82 for GPT-5, Gemini-2.5-Pro, and Qwen3-30B, respectively, and enabling all three models to reach perfect performance (F1-

Score = 1.0) at depth $D = 100$. Although the smaller Qwen3-8B model does not achieve F1-Score = 1.0 at every depth (limited to its capability), it still improves by 0.82 at depth $D = 100$, turning a previously failed setting into near-perfect performance.

### 5.2.2. FAILURE ANALYSIS

**Do LLMs degenerate into random guessing as compositional depth increases?** No, failures become increasingly structured rather than uniform across classes. We decompose the overall F1-Score trend of One-shot Reasoning (Figure 4a) by class and reveal a clear imbalance: Figure 5 shows that recall for the convex and concave classes steadily goes to 0 as depth increases, while recall for the neither class remains high. Moreover, the fraction of convex/concave functions misclassified as neither class increases with depth, indicating that errors increasingly funnel into the neither class rather than being randomly distributed.

This behavior arises because certifying convex/concave requires that convexity, monotonicity, and domain conditions hold at every level of the composition, whereas predicting neither only requires a single violation. As depth grows, local mistakes and parsing failures compound along the reasoning chain, making sharp convex/concave conclusions progressively harder to sustain. Consequently, models increasingly fall back to neither, yielding high recall for neither but a substantially degraded F1-Score.

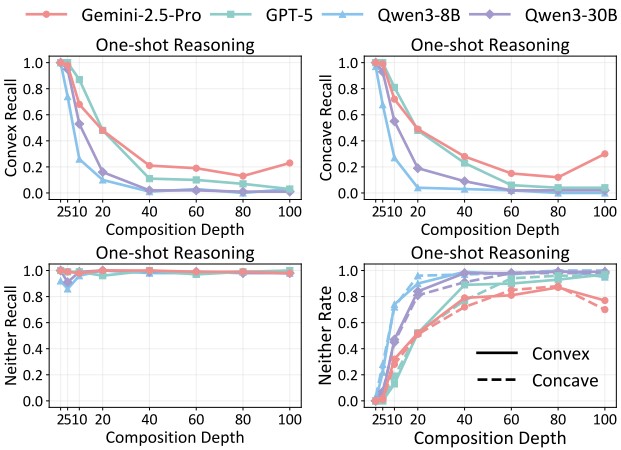

*Figure 5.* Class-wise recall for one-shot reasoning across compositional depth. Convex/concave recall degrades with depth increases, whereas neither recall remains high. Misclassifications increasingly map convex/concave inputs to neither label.

**Where does the first error mostly occur in Agentic Reasoning trace?** To analyze why the performance of Agentic Reasoning degrades at large depth, we analyze where the first error occurs along its step-by-step reasoning chain. Because it produces intermediate outputs at each step, we can identify the earliest step whose output is incorrect. Figure

6 reports the position of the first error across different compositional depths. We find that (1) as depth increases, the first error shifts toward later stages of the reasoning process, and (2) larger models (e.g., Qwen3-30B) tend to make the first mistake later than smaller ones (e.g., Qwen3-8B). Importantly, later steps are executed under a longer accumulated context with more prior intermediate states. Thus, the concentration of errors in late-stage steps is consistent with the insight that reasoning becomes more fragile as the cumulative context expands. Larger models appear more robust to this context growth, delaying the onset of the first mistake. Overall, these results support our motivation for focused context: even when step-by-step reasoning is enforced, reasoning chains can still collapse as the context grows, and pruning irrelevant history provides a principled way to reduce such late-stage errors.

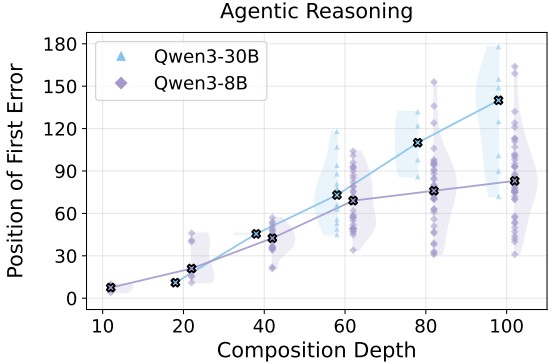

*Figure 6.* First error position of intermediate output within the Agentic Reasoning trace. As compositional depth increases, the first error consistently occurs at a later position within the reasoning trace. Moreover, smaller models are more prone to encountering their first error at an earlier position compared to larger models.

**Does compositional depth push** ConvexBench **beyond the context window of frontier LLMs?** We compute the average number of tokens for both raw and decomposed expressions across various compositional depths. As shown in Table 1, even at a depth of 100, the average token counts are only 1,912 and 5,331 for raw and decomposed expressions, respectively, far below the context limits of frontier LLMs (e.g., over 128k tokens). Nevertheless, despite operating far within the context window, models still exhibit dramatic performance degradation. This result highlights a distinction between long-context capability and long-horizon reasoning capability in current LLMs.

*Table 1.* Average token number for both raw expression and decomposed expression across different compositional depths.

| Depth | 2 | 5 | 10 | 20 | 40 | 60 | 80 | 100 |
|---|---|---|---|---|---|---|---|---|
| Raw Expression | 37 | 86 | 180 | 379 | 761 | 1148 | 1548 | 1912 |
| Decomposed Expressions | 90 | 208 | 450 | 982 | 2023 | 3087 | 4257 | 5331 |

### 5.2.3. ABLATION STUDIES AND DESIGN TRADE-OFFS

**Performance of One-shot Reasoning with Decomp across different decomposition granularities.** We investigate how the granularity of function decomposition impacts the reasoning performance of One-shot Reasoning with Decomp. Coarser decomposition consistently degrades One-shot Reasoning with Decomp. We vary the maximum sub-function length (10/50/100 characters, a proxy for sub-expression complexity in decomposition) and observe a monotonic drop in performance as sub-functions become longer (Figure 7a). A coarse decomposition produces fewer but more complex symbolic expressions, which is consistent with our earlier analysis that longer expressions reintroduce parsing and multi-condition reasoning difficulties.

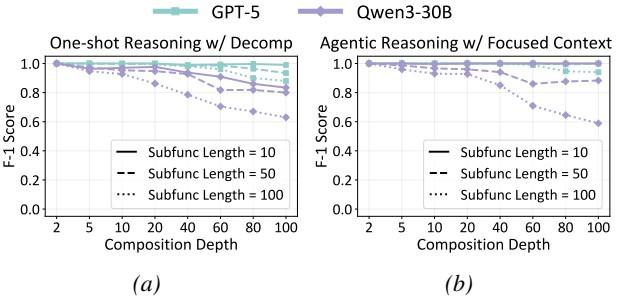

*(a)* *(b)*

*Figure 7.* Performance across different decomposition granularities. Coarser decomposition leads to increased symbolic complexity per step, resulting in a degradation of reasoning performance.

**Scaling of the recursive reasoning steps for Agentic Reasoning.** Fine-grained decomposition increases the number of recursive reasoning steps but improves performance at large depth. As shown in Table 2, at depth 100 the recursion count rises from 22 steps (length = 100) to 146 steps (length = 10). Combined with the trends in Figure 7b, this suggests that more steps with simpler local sub-expressions yield better performance. This also indicates that fine-grained decomposition and recursive reasoning can help small and mid-sized models narrow the gap on tasks that typically require more capable models.

*Table 2.* The number of Recursive reasoning steps for different decomposition granularities across different composition depths.

| Depth | 2 | 5 | 10 | 20 | 40 | 60 | 80 | 100 |
|---|---|---|---|---|---|---|---|---|
| Subfunc Length = 10 | 3 | 6 | 13 | 28 | 57 | 86 | 117 | 146 |
| Subfunc Length = 50 | 1 | 2 | 4 | 8 | 17 | 25 | 33 | 40 |
| Subfunc Length = 100 | 1 | 2 | 3 | 5 | 10 | 14 | 19 | 22 |

**Scaling of the reasoning tokens and performance-cost trade-offs.** We study how average reasoning tokens scale with composition depth under four paradigms (Figure 4e–h). Across paradigms, token scaling tracks whether the model continues to verify intermediate states as depth grows.

One-shot reasoning shows early token growth but quickly plateaus or even declines (Figure 4e), accompanied by a sharp performance drop (Figure 4a). This pattern is consistent with the lazy reasoning failure mode.

One-shot Reasoning with Decomp exhibits sub-linear token growth (Figure 4f). While decomposition boosts performance across models and keeps frontier models near the ceiling even at large depths, smaller open-source models still degrade and show slowing token growth. This suggests that decomposition mitigates parsing failures but cannot fully sustain intermediate reasoning at deep depth.

Agentic Reasoning (with Focused Context) yields near-linear token growth with depth (Figure 4h), because the procedure enforces recursive reasoning over sub-expressions. This paradigm achieves the best performance at large depths, but it incurs substantially higher token expenditure than one-shot reasoning. The benefit of these additional tokens depends on the base model. (1) For smaller models, the agentic framework acts as an essential external scaffold, transforming a previously unsolvable task into a solvable one. (2) For frontier models that are already near the ceiling on one-shot reasoning with decomposition, agentic reasoning yields limited performance gains while increasing inference costs (e.g., up to $10\times$ in our setting). One way to improve this trade-off is to batch $k$ consecutive steps into a single agentic call, instead of verifying each composition step independently. It ensures all layers are checked while reducing the number of agentic calls.

## 6. Conclusion and Discussion

Our study explores whether LLMs can recognize convex functions under deep composition and reveals a compositional reasoning gap in current LLMs. By evaluating the LLMs on ConvexBench, we find (1) parsing failure is a critical bottleneck of current LLMs in analyzing complex symbolic expressions; (2) LLMs exhibit lazy execution in long-horizon analysis; (3) performance degrades even when the input length remains far below the context limit: models condition on an expanding history of intermediate sub-functions, indicating a reasoning-horizon bottleneck beyond token-level long-context effects. Our agentic frameworks offer three practical implications. (1) When confronted with structurally complex expressions, models benefit from explicitly recognizing uncertainty and delegating parsing to external tools, an important step toward reliable automated mathematical reasoning. (2) For long-horizon analysis, recursive scaffolding improves performance over one-shot reasoning, with particularly large gains for smaller models. (3) Reasoning frameworks should be model capability-aware: for stronger models, one-shot decomposition can capture most gains at a lower cost, whereas for smaller models, decomposition combined with recursive reasoning enables progress on instances that are otherwise unsolved.

## Impact Statement

This paper presents ConvexBench, a benchmark designed to evaluate the ability of LLMs to reason about convexity under deep function compositions, and proposes an agentic framework to mitigate reasoning degradation in such settings. The goal of this work is to advance the understanding and evaluation of long-chain symbolic reasoning in machine learning models. The potential societal impact of this work is primarily positive, as it contributes to more reliable and interpretable AI systems for mathematical reasoning, optimization, and scientific modeling. Overall, we do not foresee significant negative societal consequences beyond those already well recognized in the broader use of automated reasoning systems in machine learning.

## Acknowledgment

The work of Yingbin Liang was supported in part by the U.S. National Science Foundation under the grants ECCS-2113860 and DMS-2134145.

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

# A. Examples of Failure Modes

In this appendix, we provide two examples illustrating two common failure modes observed in One-shot Reasoning (Baseline). Table 3 shows a typical parsing failure, where the model incorrectly parses an expression and makes an early structural mistake, leading to an incorrect convexity conclusion. Table 4 demonstrates lazy reasoning, in which the model relies on partial heuristics or unverified assumptions rather than performing a complete compositional analysis.

*Table 3.* An example of parsing failure in One-shot Reasoning evaluated on Qwen3-30B.

| Function | $f(x) = \exp(\exp(-(0.534(-\exp(0.656(-0.342x + 1.24) + 0.76)) + 0.238(-\|1.37x + 0.0336\|_1) - 0.263x - 1.33)))$ |
| --- | --- |
| **Abstract Syntax Tree** | $f(x) = g_8(x)$
$\quad\llcorner\; g_8(x) = \exp(\exp(-g_6(x)))$
$\qquad\llcorner\; g_6(x) = 0.534\, g_3(x) + 0.238\, g_4(x) + g_5(x)$
$\qquad\quad\llcorner\; g_3(x) = -\exp(g_2(x))$
$\qquad\qquad\llcorner\; g_2(x) = 0.656\, g_1(x) + 0.76$
$\qquad\qquad\quad\llcorner\; g_1(x) = -0.342x + 1.24$
$\qquad\quad\llcorner\; g_4(x) = -\|1.37x + 0.0336\|_1$
$\qquad\quad\llcorner\; g_5(x) = -0.263x - 1.33$ |
| **Erroneous reasoning snippet** | Let me define the innermost part. Set:
$g(x) = -\exp(0.656(-0.342x + 1.24)) + 0.76$
$\cdots$
Then, the next level is:
$h(x) = -0.534g(x) + 0.238(-\|1.37x + 0.0336\|_1) + (-0.263x - 1.33)$
$\cdots$
First, $-0.534g(x)$: $g(x)$ is concave, so negative times concave is convex (since negative of concave is convex).
Then, $0.238e(x)$: $e(x)$ is concave, so positive times concave is concave.
$\cdots$
Now, h(x) = convex part + concave part + affine part
$\cdots$
In general, sum of convex and concave is neither convex nor concave unless one dominates... |

*Table 4.* An example of lazy reasoning in One-shot Reasoning evaluated on Qwen3-30B.

| Function | $f(x) = -(0.841(0.17(0.111(0.374(1.55(-\max(0, \log(1 + \exp(0.656(0.906(0.867(\max(0, 1.08(0.165(\log(\exp(0.895$ $(\exp(\exp(-(-\exp(\log(\exp(\log(1 + \exp(\max\{\log(\exp(0.786(0.752(\exp(-(-\max(0, \max(0, \exp(-(0.945(0.279$ $(-\exp(\exp(\log(1 + \exp(\max(0, \log(1 + \exp(\exp(-(0.386(1.71(-\exp(\exp(-(-\max(0, 0.265(\max(0, \exp(-(0.664$ $(1.59(0.584(0.557(-\max(0, \max(0, \log(\exp(0.486(\max(0, -0.86x - 0.925 + \|0.471(-0.517x - 0.471)\|_2^2)) +$ $0.013(-0.823x + 1.43)) + \exp(-0.613x - 1.45))) - 0.224x + 0.757)) + 0.443(-\| - 1.21x - 1.37\|_1)) + 0.339(-\| -$ $1.14x - 2.59\|_2)) + 0.527 - \|0.474x + 2.56\|_2) + 0.203(-0.295x - 0.592))))) + 0.404(\| - 0.61x + 0.292\|_\infty))))) - \| -$ $0.857x - 1.41\|_1) + 0.435) + 0.098(-\|1.33x - 1.39\|_1) + 0.871x - 2.58)) + \| - 0.818x - 0.498\|_2))) + \| - 1.78x -$ $0.0315\|_2 - 0.542x + 1.15 + \|0.471x + 0.707\|_\infty))) + 1.03x - 0.415 + \| - 2.07x - 1.52\|_2)) + 0.721(-\|0.428x + 1.22\|_1)) +$ $0.0546(-0.115x - 0.907)))))))) - 0.786) + 0.186(-0.0526x + 0.642) + \| - 0.28x - 0.755\|_2) + \exp(-1.01x + 0.386)) +$ $1.93x + 0.427, 1.52x + 1.04\})) + \exp(\|0.225x - 0.0755\|_2)))))) - 0.462x + 0.0949) + 1.06) + \exp(\| - 0.228x -$ $0.361\|_1))) + 0.835(\|0.425x - 2.64\|_\infty) + \|0.429x - 1.43\|_1 + \|0.319x - 0.291\|_1) + 0.209 + \|0.23x - 0.0217\|_\infty + \|0.818x +$ $1.22\|_2)) + 0.133(-2.67x - 0.095)) + 0.198) + 0.344(\|0.151x + 0.793\|_\infty + \|0.943(0.778x + 0.416)\|_2^2)))) + 0.0706) +$ $0.626(-\|2.24x - 0.863\|_2)) + 0.427(-\| - 1.07x - 1.68\|_2)) + 0.825(-\|0.749x + 0.177\|_1) + 1.3x + 1.48) - 0.849)$ |
| --- | --- |
| **Erroneous reasoning snippet** | $\cdots$
The expression inside the max is convex... Then $\max(0, \text{convex})$ is convex.
$\log(1 + \exp(A))$ is convex.
$\cdots$
Wait, but let's check a part where there's a negative sign. For example, $-\max(0, \dots)$. The $\max(0, \dots)$ is convex, so negative of convex is concave.
But then, if that concave function is inside an exp, like $\exp(-\max(\dots))$, then $\exp(\text{concave})$ is not necessarily convex or concave.
For example, $g(x) = -x^2$ (concave), $\exp(g(x)) = \exp(-x^2)$, which is not convex.
$\cdots$
Therefore, $\exp(-\exp(u))$ where $u$ is convex is neither convex nor concave.
$\cdots$
Take the first term: $-(0.841 * (\dots))$.
Let's assume the inside of the $0.841 * (\dots)$ is convex (as per earlier analysis), so $0.841 * (\text{convex})$ is convex, then negative makes it concave. So the first term is concave.
$\cdots$
In the given function, the convex terms (like the norms with positive coefficients) and concave terms (norms with negative coefficients) are not canceling each other out, so the overall function will have regions where it's convex and regions where it's concave.
Therefore, the conclusion should be neither. |

# B. Experimental Details

*Table 5.* The atoms used to construct ConvexBench.

| Type | Examples |
|---|---|
| Affine | $ax + b$; |
| Convex non-decrease | $\exp(x)$; $\text{softplus}(x)$; $\text{hinge}(x)$; $\max(0, x)$; $\text{sum}(x_1, ..., x_n)$; $\max(x_1, ..., x_n)$; $\log\_\text{sum}\_\exp(x_1, ..., x_n)$ |
| Convex non-increase | $\exp(-x)$; $-\text{sqrt}(x)$, $x \geq 0$; |
| Convex non-monotonic | $\text{norm}(x, p)$, $p \geq 1$ |
| Concave non-decrease | $\log(x)$; $\text{sqrt}(x)$, $x \geq 0$ |
| Concave non-increase | $-\exp(x)$; $-\text{softplus}(x)$; $-\max(0, x)$ |
| Concave non-monotonic | $-\text{norm}(x, p)$, $p \geq 1$ |

---

**One-shot Reasoning Prompt**

### Task:
Please reason carefully step by step to determine whether the given compositional function f(x) is convex, concave, or neither.

### Procedure:
1. Decompose f(x) into its dependent sub-functions G_i.
2. Identify dependent sub-functions used in G_k. For each dependent sub-function G_i, you must check and fully use all of:
    (1) expression form (affine, exp(·), log(·), max, sum, etc.)
    (2) curvature (affine/convex/concave/neither)
    (3) range (pos/neg/any)
3. Apply DCP rules, for example:
    (1) sum of convex → convex; sum of concave → concave;
    (2) negative of convex → concave; negative of concave → convex;
    (3) max(convex, convex) → convex; min(concave, concave) → concave;
    (4) If u is convex or affine, then exp(u) → convex;
    (5) if u is concave or affine, then log(u) → concave;
    (6) log-sum-exp(u_i) is convex if each u_i is convex/affine: For an outer function log(...), e.g., log(G_1 + G_2), you must check each dependent G_i (expression form, curvature, and range). You may apply the log-sum-exp rule only when every term is explicitly of the form exp(u_i) and each u_i is convex/affine. If any term is not of the form exp(u_i), then you cannot apply the log-sum-exp rule. For example: Assume G_3 = log(G_1 + G_2) is convex on its domain, G_4 = exp(0.3x + 0.5) is convex, then G_5 = log(exp(G_3) + G_4) = log(exp(G_3) + exp(0.3x + 0.5)) is convex by log-sum-exp;
    (7) log(1+exp(u)) is convex if u is convex/affine; -log(1+exp(u)) is concave if u is convex/affine

### Output format (Only this JSON format, nothing else):
{
    "analysis": "A clear, detailed, step-by-step explanation of your reasoning that leads to your conclusion",
    "conclusion": "convex | concave | neither"
}

Now, analyze the following function:

---

**Agentic Reasoning Prompt**

### You are given:
(1) A list of subfunctions with their names, expressions, curvature annotations, and ranges.
(2) A target subfunction G_k to analyze.

### Goal:

Please reason carefully step-by-step to determine the curvature (affine | convex | concave | neither) and range (pos | neg | any) of G_k.

### Procedure:
1. Identify dependent sub-functions used in G_k. For each dependent sub-function G_i, you must check and fully use all of:

    (1) expression form (affine, exp(·), log(·), max, sum, etc.)

    (2) curvature (affine/convex/concave/neither)

    (3) range (pos/neg/any)

2. Apply DCP rules, for example:

    (1) sum of convex → convex; sum of concave → concave;

    (2) negative of convex → concave; negative of concave → convex;

    (3) max(convex, convex) → convex; min(concave, concave) → concave;

    (4) If u is convex or affine, then exp(u) → convex;

    (5) if u is concave or affine, then log(u) → concave;

    (6) log-sum-exp(u_i) is convex if each u_i is convex/affine: For an outer function log(...), e.g., log(G_1 + G_2), you must check each dependent G_i (expression form, curvature, and range). You may apply the log-sum-exp rule only when every term is explicitly of the form exp(u_i) and each u_i is convex/affine. If any term is not of the form exp(u_i), then you cannot apply the log-sum-exp rule. For example: Assume G_3 = log(G_1 + G_2) is convex on its domain, G_4 = exp(0.3x + 0.5) is convex, then G_5 = log(exp(G_3) + G_4) = log(exp(G_3) + exp(0.3x + 0.5)) is convex by log-sum-exp;

    (7) log(1+exp(u)) is convex if u is convex/affine; -log(1+exp(u)) is concave if u is convex/affine

### Output format (Only this JSON format, nothing else):
{

    "curvature": "affine | convex | concave | neither",

    "range": "pos | neg | any"

}

Now, analyze the following function:

