# OpenReview forum: "ConvexBench: Can LLMs Recognize Convex Functions?"
_ICML.cc/2026/Conference — ICML 2026 regular_

### Official Review · Reviewer_UBBU · 2026-02-16

**Soundness:** 3
**Presentation:** 3
**Significance:** 2
**Originality:** 2
**Overall Recommendation:** 5
**Confidence:** 3

**Summary:**

This work introduces a language-based text dataset designed to evaluate the capability of LLMs to recognize whether a function is convex, concave, or neither. Experiments have been conducted using several instruction-tuned LLMs from the Google Gemini family, OpenAI models, and the Alibaba Group Qwen family.

**Compliance With Llm Reviewing Policy:**

Affirmed.

**Final Justification:**

I increased my score from weak accept to accept. See my comments below.

**Key Questions For Authors:**

1. Regarding the Jensen’s test, by this, do you mean the inequality involving expectation, $(f(\mathbb{E}[X]) \leq \mathbb{E}[f(X)])$? If so, which distributions did you choose for testing? Did you encounter any cases where the inequality holds for a specific distribution but the function does not satisfy it in general? Alternatively, did you use the discrete form of Jensen’s inequality, $(\sum_i \lambda_i x_i \leq \sum_i \lambda_i f(x_i))$ with $(\lambda_i \ge 0)$ and $(\sum_i \lambda_i = 1)$?

2. What exact information does the agent provide? Does the agent add only a hint to the prompt, or, in the reasoning process, does it focus on a particular part of the function? For example, if we have $(f_3(f_2(f_1)))$, does the hint for $(f_2)$ instruct the model to consider only $(f_2)$ and $(f_1)$ while ignoring $(f_3)$?

3. In line 302, when you refer to “tokens,” do you mean the number of words or the number of tokens as output by each model’s tokenizer? Additionally, in line 310, you mention that the maximum number of output tokens is 50,000. What were the minimum, maximum, and average number of tokens actually produced by the models?

**Limitations:**

Including a dedicated Limitations section after the Conclusion would significantly strengthen the paper.

**Strengths And Weaknesses:**

Strengths:

1. Overall, this is a well-executed paper. The methodology is clearly explained and well motivated, and the central concept of mathematical reasoning is both interesting and meaningful. The way the authors construct the dataset and evaluate the models is thoughtful and systematic.

Main Weaknesses:

2. There is no training phase involving open-weight LLMs on the proposed dataset to evaluate whether these models can effectively learn and strengthen the targeted reasoning abilities. As the work is primarily dataset-focused, it would be beneficial to include experiments analyzing the training dynamics and adaptation of models on this dataset, which would provide stronger evidence of its practical utility.

Minor Comments:

3. There are several typographical and formatting issues that should be corrected. For example, abbreviations should follow consistent capitalization (e.g., “Large Language Models (LLMs)” rather than “large language models (LLMs)”). Similarly, “Disciplined Convex Programming (DCP)” in line 52 should use proper capitalization. In addition, paragraph and section titles should follow consistent capitalization, such as “Our Contributions” (line 107), “LLMs for Mathematical Reasoning” (line 120), and “Agentic Reasoning with Focused Context Closes the Compositional Reasoning Gap” (line 321).

4. Several citations are listed only as arXiv preprints. For works that have appeared in conferences or journals, the officially published versions should be cited instead to maintain citation quality and formality.

5. The citation for Qwen3-8B in line 325 appears to be incorrect, as it references DeepSeek rather than the intended model. Please verify and replace it with the correct official citation (https://huggingface.co/Qwen/Qwen3-8B).

---

> ### Author Rebuttal · Authors · 2026-03-31
>
> We appreciate your constructive feedback and are pleased you found our benchmark systematic and our method meaningful. Our detailed responses follow.
> ## W1. Training dynamics of models on dataset.
> We agree that it is interesting to study whether training on ConvexBench can help models internalize compositional reasoning, and we will add this as a discussion of future direction in the revised manuscript. At the same time, we view this as a substantial project of its own, involving training dynamics, generalization, and the risk of overfitting to DCP patterns. As such, it naturally builds on, rather than replaces, the evaluation and inference-time contributions of the present work.
>
> Moreover, we clarify the focus of this work: 1) evaluating LLMs' compositional reasoning capabilities, 2) proposing targeted methods that both verify the identified failure modes and mitigate them at inference time.
> ## W2. Typographical and formatting issues.
> We will fix those inconsistencies in the revised manuscript.
> ## W3. Citations format.
> We will check and update all citations to their published versions.
> ## W4. Citation of Qwen3-8B.
> We clarify that the citation is actually correct, the model we used is DeepSeek-R1-Qwen3-8B. We will rename it to DS-Qwen3-8B to avoid confusion.
> ## Q1. Clarification of Jensen’s test.
> In our implementation, the Jensen test is discrete finite-sample form, i.e., if $f(a x+(1-a)y) \le f(x)+(1-a)f(y)$.  $x,y$ are sampled from a Gaussian distribution.
>
> Moreover, Jensen's test serves two distinct roles depending on the target class:
> - For convex/concave targets: Convexity is already guaranteed by DCP rules. Jensen's test is only a sanity check not the source of the label.
> - For neither class: Jensen's test acts as a counterexample filter. We admit a function as neither only if we find violations of both the convexity and concavity.
>
> Regarding your concern of false negatives: this applies only to neither class. Our design handles it conservatively: if we fail to find a counterexample, we reject the candidate.
>
> ## Q2. Implementation details of the agentic framework
> The agent does not add hints to the prompt telling the model what to focus on. Instead, it uses external scaffolding to structurally control what information the LLM receives at each reasoning step. The model focuses on a particular part of the function because that is the only part presented to it.
> ## Q3. Term clarification and statistics of reasoning token number
> - Clarification of the term tokens
>
>     The tokens reported in our paper refer to tokens as counted by each model's respective tokenizer, not word counts.
>
> - Min, max and average number of reasoning tokens
>
>     We clarify that the average number of reasoning tokens for different methods are presented in Figure 4. We compute the minimum and maximum number of tokens in two tables below.
>
> Table1: Minimum number of tokens. A:GPT-5, B: Gemini-2.5-pro, C: Qwen3-8B, D:Qwen3-30B. OR: One-shot reasoning, ORD: One-shot reasoning w/ decomp, AG: Agentic reasoning, AGFC: Agentic reasoning w/ focused context.
> |Methods|Models|2|5|10|20|40|60|80|100|
> |-|-|-|-|-|-|-|-|-|-|
> |OR|A|960|2112|6912|13248|10496|9152|10496|8768|
> ||B|810|1946|3429|3406|3035|2800|3424|3854|
> ||C|767|2366|3570|4639|4179|4088|3939|3620|
> ||D|371|745|2079|3447|2823|2401|2423|2464|
> |ORD|A|448|2240|4608|5312|6464|8768|6528|8640|
> ||B|810|1128|1775|3078|3927|3443|3920|3209|
> ||C|590|1103|2160|2777|3488|3792|4146|4020|
> ||D|401|868|1284|1933|1840|1605|1625|1341|
> |AG|A|896|6336|16064|64384|136832|219904|301312|395648|
> ||B|603|2688|5703|15742|31734|49417|78892|82735|
> ||C|2491|6478|23261|57110|143609|266758|420762|372220|
> ||D|853|3789|8517|22662|49613|79352|112721|134849|
> |AGFC|A|896|6272|21120|69056|134592|236224|351360|417344|
> ||B|723|2468|7904|18269|38525|56497|80480|108235|
> ||C|1969|7533|20689|18828|131640|244760|345853|368676|
> ||D|761|3285|9179|22967|56955|91940|131762|149448|
>
> Table2: Maximum number of tokens
> |Methods|Models|2|5|10|20|40|60|80|100|
> |-|-|-|-|-|-|-|-|-|-|
> |OR|A|15872|23680|36032|46656|40896|49728|34176|39872|
> ||B|10187|11183|15190|18139|19334|21515|21110|23182|
> ||C|12923|11599|86288|20112|78467|41936|67938|69786|
> ||D|13818|16008|13794|13893|13211|14677|11855|17785|
> |ORD|A|16448|14976|18432|20544|21440|25792|27776|33408|
> ||B|10187|11537|10178|8560|13924|16903|21490|26103|
> ||C|13854|12830|10577|12845|15201|18890|39042|20327|
> ||D|14773|10085|11357|9951|11978|14351|15699|30776|
> |AG|A|13120|28096|64960|126144|228096|273792|372608|454912|
> ||B|5533|9053|13876|29313|51092|76111|100691|123114|
> ||C|18561|28806|60269|131357|277157|434696|533091|742845|
> ||D|13887|13410|23214|51892|92348|137600|182822|243890|
> |AGFC|A|11840|21184|53120|97280|217920|254720|381376|454976|
> ||B|5905|7685|12915|33375|57028|77449|115644|142312|
> ||C|17283|24382|57830|233871|221717|334607|440583|544310|
> ||D|10345|13079|25568|62108|90125|138776|188726|237374|
>
> ## Q4. Include a Limitations section
> We will add a Limitations section in the revised manuscript.

---

> > ### Author Rebuttal · Reviewer_UBBU · 2026-03-31
> >
> > I would like to thank the authors for their detailed and helpful response.
> >
> > I have a couple of clarification questions to ensure that I correctly understand the methodology.
> >
> > Regarding Q1:
> > From your explanation, my understanding is the following:
> >
> > For convex/concave labels:
> >   * The source of truth is the DCP rules.
> >   * Jensen’s test is used only as a sanity check, not as the primary criterion for assigning the label.
> >
> > For the “neither” label:
> >   * The source of truth is the presence of Jensen violations.
> >   * Jensen’s test is used as the main decision tool, ie, a function is classified as neither convex nor concave only if counterexamples to both properties are found.
> >
> > Could you please confirm whether this interpretation is correct?
> >
> >
> > Regarding Q2:
> >
> > I would also like to clarify my understanding of the agent’s role. For example, given a composed function such as ( f_3(f_2(f_1(x))) ), is the process roughly as follows?
> >
> > 1. The agent first provides ( f_1 ) to the model and the model analyzes it.
> > 2. Then, the agent provides ( f_2 ), possibly using the result or abstraction of ( f_1 ).
> > 3. Finally, the agent provides ( f_3 ), and the model analyzes it in the same manner.
> >
> > In other words, at each step, the LLM only evaluates the currently provided function component, rather than reasoning over the entire composition at once.
> >
> > Could you please confirm if this description accurately reflects the agent’s behavior?

---

> > > ### Author Response · Authors · 2026-03-31
> > >
> > > Dear Reviewer UBBU,
> > >
> > > Thank you very much for your follow-up questions and for the positive feedback on our previous response.
> > >
> > > We confirm that your interpretations of both our labeling process (Q1) and the agentic framework's implementation (Q2) are entirely correct. We appreciate your clear and accurate synthesis of these points, and we will ensure they are explicitly reflected in the revised manuscript.
> > >
> > > Best regards,
> > >
> > > Paper13296 Authors

---

### Official Review · Reviewer_ic9U · 2026-03-11

**Soundness:** 3
**Presentation:** 3
**Significance:** 2
**Originality:** 2
**Overall Recommendation:** 4
**Confidence:** 3

**Summary:**

This paper studies whether large language models (LLMs) can correctly identify the convexity of functions under deep functional composition. To systematically evaluate this capability, the authors introduce ConvexBench, a scalable and mechanically verifiable benchmark that generates symbolic objective functions with varying composition depths and asks models to classify them as convex, concave, or neither. Through experiments on several frontier and open-source LLMs, the study reveals a significant compositional reasoning gap: model performance drops sharply as the depth of functional composition increases, even when the total input length remains well within the models’ context window.
The authors further analyze the causes of these failures and identify two primary failure modes: parsing failures when handling complex symbolic expressions and lazy reasoning, where models fail to sustain consistent intermediate reasoning across long reasoning chains. To address these limitations, the paper proposes an agentic divide-and-conquer framework that delegates expression parsing to an external tool to construct an abstract syntax tree (AST) and enforces recursive reasoning over intermediate sub-expressions with focused context. Experiments show that this approach substantially improves performance at large composition depths and can recover near-perfect classification accuracy even in very deep compositions. The paper also includes ablation studies examining decomposition granularity, reasoning token scaling, and the trade-offs between reasoning cost and performance.

**Compliance With Llm Reviewing Policy:**

Affirmed.

**Final Justification:**

The rebuttal has addressed my main concerns and I would like to raise my score to 4

**Key Questions For Authors:**

1.	Generality of ConvexBench beyond convexity reasoning.
ConvexBench focuses on convexity classification under deep symbolic composition. To what extent do the authors believe that the identified failure modes (e.g., parsing failure and lazy reasoning) generalize to other symbolic reasoning or mathematical reasoning tasks? For example, would similar performance degradation appear in tasks such as symbolic algebra, program analysis, or theorem-style reasoning? Clarifying this would help better understand the broader significance of the benchmark. If the authors can provide evidence that these failure modes generalize to other reasoning tasks, it would strengthen the significance of the work.

2.	Dependence on external parsing tools.
The proposed agentic framework relies on an external tool to construct the abstract syntax tree (AST). Could the authors clarify how sensitive the overall performance is to the quality or design of this parser? For example, would different parsing strategies significantly affect the results? If the approach strongly depends on a specific parser implementation, this may limit the general applicability of the framework.

3.	Cost–performance trade-off of the agentic framework.
The experiments show that the agentic reasoning framework significantly increases token usage compared with one-shot reasoning. Could the authors provide a more detailed analysis of the computational cost and practical deployment considerations? For example, how does the method scale in terms of inference latency or cost for real-world applications? A clearer characterization of this trade-off could help readers evaluate the practical usefulness of the proposed method.

4.	Robustness across model families and sizes.
The experiments evaluate a limited number of LLMs. Have the authors tested whether the observed compositional reasoning gap is consistent across a wider range of models or architectures? If similar behavior appears across many model families, it would strengthen the claim that this is a fundamental limitation of current LLM reasoning.

**Limitations:**

yes

**Strengths And Weaknesses:**

Soundness:
o	Strengths: The paper proposes a mechanically verifiable benchmark (ConvexBench) and evaluates multiple LLMs across a wide range of compositional depths, which provides a controlled and systematic way to study compositional reasoning. The experiments include useful analyses such as failure mode identification and ablation studies, which help support the main claims.
o	Weaknesses: The empirical evaluation is somewhat limited. Only a small number of models are evaluated, and the conclusions about reasoning limitations may therefore be somewhat narrow. Additionally, the agentic framework increases inference cost, yet the analysis of the cost–performance trade-off is relatively limited and does not fully justify the practicality of the proposed method.

Presentation:
o	 Weaknesses: Although the paper is generally readable, some methodological components are not described with sufficient clarity. In particular, the interaction between the external parsing tool and the recursive reasoning process could be explained more concretely. The paper would benefit from clearer algorithmic descriptions (e.g., pseudocode) to improve reproducibility.
Significance:
o	Weaknesses: The task studied in the paper—convexity classification under symbolic composition—is quite narrow and specialized. While it is an interesting diagnostic setting, it is not entirely clear how well the findings generalize to broader reasoning tasks or real-world applications of LLMs.

Originality:
o	Strengths: The introduction of ConvexBench provides a new benchmark targeting compositional symbolic reasoning in convex analysis, which fills a gap in existing evaluation settings.
Weaknesses: The proposed agentic reasoning framework mainly combines existing ideas such as decomposition, tool use, and recursive reasoning. As a result, the methodological novelty is somewhat limited, and the contribution is more incremental in nature.

---

> ### Author Rebuttal · Authors · 2026-03-31
>
> Thank you for recognizing ConvexBench as a critical, mechanically verifiable benchmark for compositional reasoning. For your valuable reviews, we provide responses below.
>
> ## W1. Parser-reasoning interaction and algorithmic pseudocode
>
> The external parser is used as an initial step to deterministically decompose symbolic expressions into a topologically ordered sequence of sub-functions. Our recursive reasoning executes over this structured decomposition within the proposed scaffold. We provide detailed pseudocodes below.
>
> ```cpp
> A1: One-shot Reasoning w/ Decomp
>
> INPUT: LLM M(), expression F(), parser D()
>
> 1: G = {g_1, g_2, ..., g_n} <- D(F(x))
> 2: y <- M(G)
>
> OUTPUT: y
> ```
> ```cpp
> A2: Agentic Reasoning
>
> INPUT: LLM M(), expression F(), parser D()
>
> 1: G = {g_1, g_2, ..., g_n} <- D(F(x))
> 2: σ_1 = M(g_1)
> 3: for i = 2,...,|G| do
> 4:	C_i <- {(g_j, σ_j) | 1 <= j < i}
> 5:	σ_i <- M(C_i, g_i)
> 6: y <- σ_{|G|}
>
> OUTPUT: y
> ```
> ```cpp
> A3: Agentic Reasoning w/ Focused Context
>
> INPUT: LLM M(), expression F(), parser D()
>
> 1: G = {g_1, g_2, ..., g_n} <- D(F(x))
> 2: σ_1 <- M(g_1)
> 3: for i = 2,...,|G| do
> 4:	C_i <- {(g_j, σ_j) | j ∈ Pa(i) }
> 5:	σ_i <- M(C_i, g_i)
> 6: y <- σ_{|G|}
>
> OUTPUT: y
> ```
>
> ## W2. Clarify the contribution of proposed methods.
>
> **We clarify that the contribution of proposed methods is not simply decomposition or tool use. Rather, it reveals a specific failure mode that current LLMs still don’t reliably overcome, even though these strategies are available in principle.**
>
> Specifically, 1) rather than a heuristic combination of existing components, our method is a diagnosis-driven design rooted in a systematic analysis of failure modes; 2) the significant performance gains further validate the identified failure modes; 3) our approach provides insights into how LLMs can overcome lazy reasoning and structural parsing challenges.
>
> ## Q1. Generalization of failure modes to other reasoning tasks.
>
> **Our view is that these failures are driven less by the specific concept of convexity, and more by task structure.**
>
> In ConvexBench, **the difficult part is not advanced convex analysis**: each local step is elementary and can be solved by applying DCP rules. **The challenge is that the model must recover the correct symbolic structure, and reliably propagate local properties through a deep dependency chain, where any intermediate mistake fails conclusions.**
>
> These failure modes extend beyond convexity: (1) parsing failure corresponds to failing to reconstruct the latent dependency structure, (2) lazy reasoning reflects a failure in local-to-global rule propagation. The same two requirements arise in other symbolic tasks such as algebraic manipulation.
>
> ## Q2. Sensitivity to parser quality and design
>
> While our framework requires an accurate parser, its performance gains are agnostic to specific implementation. The parser deterministically exposes symbolic dependencies, isolating parsing failures found in one-shot settings. In Figure 7, sensitivity lies in the granularity of decomposition: performance drops as sub-functions become structurally less explicit. Thus, any parser faithfully extracting the dependency for recursive verification is suitable.
>
> ## Q3. Cost–performance trade-off of the agentic framework
>
> We provided a detailed cost-performance analysis in Section 5.2.3. Figure 4 indicate that one-shot reasoning with decomposition significantly boosts performance, especially for proprietary models, with negligible extra token costs. While agentic reasoning achieves a perfect F1 score, it incurs a $~20\times$ token increase. Latency (/s) for GPT-5 and Qwen3-30B are provided below.
>
> This trade-off depends on model capability: for frontier models, one-shot reasoning w/ decomposition achieves great performance, rendering agentic framework redundant. However, the agentic scaffold is indispensable for smaller models to transform unsolvable tasks into solvable ones.
>
> Table1 (OR: One-shot reasoning, ORD: One-shot reasoning w/ decomp, AG: Agentic reasoning, AGFC: Agentic reasoning w/ focused context)
> |Model|Method|2|5|10|20|40|60|80|100|
> |-|-|-|-|-|-|-|-|-|-|
> |GPT-5|OR|124.32|165.35|255.58|413.44|405.97|360.25|353.23|325.22|
> ||ORD|112.33|142.47|193.45|205.62|248.65|303.87|348.91|377.65|
> ||AG|106.76|243.83|650.78|1502.21|2862.28|4259.17|5688.10|7246.31|
> ||AGFC|107.95|264.23|594.68|1532.29|3269.75|4437.93|6312.35|6783.75|
> |Qwen3-30B|OR|59.63|71.52|92.62|139.05|334.56|361.92|435.54|390.87|
> ||ORD|50.94|52.95|55.69|68.22|98.24|117.17|145.86|165.11|
> ||AG|65.00|129.80|269.84|561.11|1151.57|1774.50|2013.38|3271.94|
> ||AGFC|63.85|112.20|238.30|557.54|1140.92|1659.35|2349.91|2818.31|
>
> ## Q4. Generalization of the reasoning gap across models
>
> We reproduced the gap on a different model family NVIDIA Nemotron3-30B. In table below, the consistent degradation confirms that the gap is a fundamental limitation instead of a model-specific artifact.
>
> |Depth|2|5|10|20|40|60|80|100|
> |-|-|-|-|-|-|-|-|-|
> |F1|1.00|0.98|0.79|0.29|0.14|0.17|0.16|0.16|

---

> > ### Author Rebuttal · Reviewer_ic9U · 2026-04-03
> >
> > Thank you for the detailed response and the additional experiments, which have significantly improved the empirical support of the work. The revised evaluation is now more comprehensive and better substantiates the effectiveness of the proposed framework, making the overall contribution clearer and more convincing.
> >
> > While I still have some reservations regarding the level of methodological novelty—since several components build upon established techniques—and the generalizability beyond symbolic domains remains somewhat unclear, these concerns are secondary to the now solid experimental validation.
> >
> > Given these improvements, I am willing to raise my score to 4.

---

> > > ### Author Response · Authors · 2026-04-03
> > >
> > > Dear Reviewer ic9U,
> > >
> > > Thank you very much for your thoughtful reconsideration and for raising the score. We are encouraged that our additional experiments and revised evaluation have addressed your concerns regarding the empirical support and the effectiveness of the proposed framework.
> > >
> > > Regarding the remaining reservations, we appreciate your insights on methodology and generalizability. We will incorporate a discussion of these perspectives in the final version to provide a more balanced and comprehensive view of our work. We will also further explore the generalizability in our future work.
> > >
> > > Thank you again for your constructive guidance throughout the review process, which has significantly strengthened our work.
> > >
> > > Best regards,
> > >
> > > Paper13296 Authors

---

### Official Review · Reviewer_ajtm · 2026-03-11

**Soundness:** 3
**Presentation:** 3
**Significance:** 2
**Originality:** 3
**Overall Recommendation:** 5
**Confidence:** 3

**Summary:**

This paper test the LLMs ability in math reasoning in a specific domain: recogonizing convex functions. To be specific, in this paper, convex functions are synthesized through a recursive composition process. Then LLMs are tested whether they can correctly identify convex functions. The results showed that when the composition depth increase, the performance dropped dramatically. The authors then proposed a agentic divide-and-conquer framework to improve the performance

**Compliance With Llm Reviewing Policy:**

Affirmed.

**Final Justification:**

I will keep my original score 5 (accept).

**Key Questions For Authors:**

I am wondering that, it seems that deciding convexity is NP hard for general polynomial representations. So is this a good task for LLMs? Why can you hope that LLM can solve this kind of problem?

**Limitations:**

Yes

**Strengths And Weaknesses:**

Strengths:

Originality: The paper introduces a novel benchmark for evaluating whether LLMs can recognize convexity under deep function composition. This is a fresh and well-motivated problem setting: rather than evaluating mathematical reasoning in a broad and potentially noisy way, the paper focuses on a structured compositional reasoning task with mechanically verifiable labels.

Soundness: The paper is empirically solid overall. The benchmark construction is carefully designed, and the labels are derived through rule-based generation and verification rather than subjective human annotation. The experiments are systematic across different composition depths and model families, and the ablation studies provide useful support for the paper’s main claims.

Presentation: The paper is generally well written and easy to follow. The problem formulation, benchmark design, and experimental setup are clearly presented, and the overall narrative is coherent. The paper also does a good job connecting the observed failure modes with the proposed method.

Weaknesses:

Significance: Although the benchmark is novel, its broader significance is still somewhat unclear. In particular, recognizing convexity for deeply composed functions may be a challenging task even for humans without access to external symbolic tools, which raises some question about the practical relevance of the setting. In addition, the proposed agentic method appears to derive much of its benefit from external parsing and decomposition tools, so part of the improvement may come from simplifying the task structure rather than advancing the model’s intrinsic reasoning ability. As a result, the overall contribution may feel more like an effective engineering scaffold than a substantial step forward in mathematical reasoning.

---

> ### Author Rebuttal · Authors · 2026-03-31
>
> Thank you for your constructive and positive comments. We sincerely appreciate your recognition of ConvexBench’s novelty and the empirical soundness of our systematic evaluation. For your valuable reviews, we provide our detailed responses below.
>
> ## Q1. Practical Relevance of the Benchmark and the Contribution of the Proposed Scaffold to Mathematical Reasoning.
>
> **1. On practical relevance.**
> We believe the difficulty of human convexity reasoning reinforces, rather than undermines, our motivation.
>
> - **One intention of ConvexBench is to evaluate long-horizon symbolic reasoning capability rather than domain-specific difficulty.** The task doesn’t require advanced convex analysis; instead, it relies on the propagation of DCP rules. Checking the DCP rules step by step is tedious for humans, and it is well suited to being handled by AI. The core challenge lies in managing long-chain dependencies and preventing error propagation over deep compositions.
>
> - **Practical value of convexity identification.** In practice, complex objectives are routinely constructed through modular reuse and repeated transformations, and verifying their convexity is essential for selecting appropriate optimization algorithms.  If LLMs could reliably perform this analysis directly from symbolic expressions, it would meaningfully lower the barrier for practitioners. ConvexBench is designed to measure progress toward this goal.
>
>
> **2. On external scaffold and mathematical reasoning.**
>
> **We clarify that agentic scaffolding is not in opposition to model's mathematical reasoning, but a crucial component for advancing it.** Designing effective external scaffolding to unlock LLMs' reasoning potential is itself a recognized research direction. For example, [1] demonstrated that a model-agnostic verification-and-refinement scaffold can boost IMO 2025 accuracy from ~30% to ~85.7%, a gain derived entirely from structured scaffolding. Our work follows this philosophy.
>
> Moreover, our framework reveals some nature of the failures. The fact that the same models achieve F1=1.0 under our framework demonstrates that they already possess sufficient DCP knowledge. They fail not due to a knowledge deficit but due to an inability to organize multi-step reasoning over complex structures. Identifying this distinction is also a contribution to understanding LLM reasoning.
>
> [1] Huang, Yichen, and Lin F. Yang. "Winning Gold at IMO 2025 with a Model-Agnostic Verification-and-Refinement Pipeline." arXiv preprint arXiv:2507.15855.
>
>
>
> ## Q2. General convexity testing is NP-hard. Why expect LLMs to solve this?
>
>
> We clarify that our benchmark doesn't target general convexity testing for arbitrary symbolic expressions. Instead, (1) Every instance is strictly restricted to DCP-style atoms and certified composition rules. This ensures that the reasoning step remains elementary and can be solved in linear time. (2) The difficulty lies not in advanced convex analysis, but in the reliable propagation of properties through deep composition chains.

---

> > ### Author Rebuttal · Reviewer_ajtm · 2026-04-05
> >
> > My concerns are fully resolved by the authors' response.

---

> > > ### Author Response · Authors · 2026-04-05
> > >
> > > Dear Reviewer ajtm,
> > >
> > > Thank you very much for your positive feedback and for acknowledging our rebuttal. We are glad to hear that our responses have fully addressed your concerns. We will further improve the manuscript based on your valuable comments in the final version.
> > >
> > > Best Regards,
> > >
> > > Paper13296 Authors

---

### Official Review · Reviewer_fvkS · 2026-03-12

**Soundness:** 3
**Presentation:** 3
**Significance:** 2
**Originality:** 4
**Overall Recommendation:** 5
**Confidence:** 4

**Summary:**

This paper studies whether LLMs can reliably determine the convexity of symbolic functions as compositional depth increases. To support this question, the authors introduce ConvexBench, a controllable and mechanically verifiable benchmark for convexity identification, and use it to evaluate how model performance changes from shallow to deeply nested expressions. The paper shows that standard one-shot reasoning degrades substantially as depth grows, and suggests that this trend is largely driven by parsing failures and lazy reasoning. To address these issues, the authors propose reasoning frameworks based on external decomposition, recursive intermediate verification, and focused context management. Experiments on several frontier models suggest that these structured reasoning strategies can substantially improve performance on deep compositions compared with one-shot prompting.

**Compliance With Llm Reviewing Policy:**

Affirmed.

**Final Justification:**

The author explained it very clearly. I think this article is a piece worth reading.

**Key Questions For Authors:**

**1. Could the authors provide a more controlled comparison under matched inference budgets?**

It is still hard to tell how much of the gain comes from better reasoning structure versus higher inference cost, especially given the different sampling settings across model families. A matched-budget comparison, such as fixed total tokens or fixed numbers of model calls, would increase my confidence in both the soundness and the practical value of the reported gains.

**2. How should the reported gains be interpreted: as improvements in model reasoning, or as improvements from external scaffolding?**

The proposed system relies on external parsing, explicit decomposition, recursive verification, and focused context management. A clearer framing of what is being measured here would help me assess the paper’s contribution and position it more precisely relative to prior work.

**Limitations:**

yes

**Strengths And Weaknesses:**

The paper is technically reasonable and empirically convincing in its core benchmark design and experimental narrative. A clear strength is that ConvexBench is built with mechanically verifiable labels and a controllable notion of compositional depth, which makes the evaluation cleaner and more reliable than many reasoning benchmarks with ambiguous supervision. The empirical results also support the paper’s main claim well: one-shot reasoning degrades substantially as depth increases, and the proposed decomposition and recursive reasoning strategies lead to large improvements, especially the focused-context variant. The paper further strengthens this case by providing concrete failure-mode analysis centered on parsing failures and lazy reasoning, and these diagnoses are well aligned with the proposed method design.

My main weakness on soundness is that the evaluation could be more rigorous in terms of cost comparability and controlled budget matching across reasoning paradigms. The paper shows clear gains from decomposition and agentic reasoning, but the strongest improvements come with substantially higher token usage and more structured external scaffolding. In addition, the paper uses different sampling settings across model families, with self-consistency for open-source models and single-sample evaluation for proprietary models. As a result, it is not yet fully clear how much of the gain should be attributed to better reasoning structure itself, versus increased inference budget and system-level support. This does not invalidate the main result, but the empirical case would be stronger with more explicit matched-budget comparisons, such as fixed total tokens, fixed numbers of model calls, or more controlled comparisons between decomposition-only and full agentic variants.

In terms of presentation, I found the paper generally clear, well organized, and easy to follow. The benchmark motivation, method progression, and experimental story are presented in a coherent order, and I do not see major issues with readability. My only minor reservation is that some claims could more clearly distinguish results established on ConvexBench from broader implications about long-horizon symbolic reasoning more generally.

I view the paper as significant in a meaningful, if somewhat specialized, way. The question it studies is important: whether LLMs can sustain correct reasoning over deep compositional chains, even when the underlying local rules are simple. Convexity identification provides a clean and verifiable setting for studying this issue, and the resulting benchmark seems useful as a diagnostic tool for long-horizon symbolic reasoning. The paper’s findings may therefore be valuable both for evaluating reasoning systems and for motivating scaffolded or agentic approaches in structured mathematical settings.

In terms of originality, I think the strongest contribution is the benchmark and the empirical perspective, rather than the method itself. ConvexBench provides a reasonably novel and well-motivated testbed for studying compositional symbolic reasoning under controlled depth. The paper also offers useful insight by separating performance degradation into parsing-related and reasoning-related failure modes. By contrast, the method is a sensible and effective combination of existing ideas—external parsing, decomposition into subproblems, recursive verification, and dependency-focused context management—rather than a fundamentally new reasoning framework. I therefore see the originality as moderate but meaningful overall.

---

> ### Author Rebuttal · Authors · 2026-03-31
>
> Thank you for your constructive feedback, and we are encouraged by the positive recognition of ConvexBench’s design and our framework's effectiveness. We highly value your review and have carefully discussed your comments below.
>
> ## Q1. Provide a controlled comparison under matched inference budgets
>
> **1. Where does the performance gain come from: better reasoning structure or higher inference cost?**
>
> The answer is both, and importantly, neither alone is sufficient. We conduct two controlled experiments on Qwen3-30B to disentangle these factors.
>
> **Experiment 1: Increasing inference budget alone does not help.** We scale the self-consistency sampling with majority vote from N=64 to N=256 for one-shot reasoning (with and without decomposition). As shown in Table 1, performance remains essentially unchanged across all depths, demonstrating that simply spending more inference budget without structural improvement yields no meaningful gain.
>
> Table 1: One-shot reasoning (with and without decomposition) on different self-consistency sampling
> | Methods  | Self-consistency sampling | 2   | 5  | 10 | 20 | 40 | 60 | 80 | 100|
> |--------------------------------|---------------------------|------|------|------|------|------|------|------|------|
> | One-shot Reasoning             | N = 64                    | 1.00 | 0.93 | 0.69 | 0.38 | 0.24 | 0.19 | 0.18 | 0.18 |
> |                                | N = 128                   | 1.00 | 0.94 | 0.71 | 0.39 | 0.23 | 0.17 | 0.19 | 0.21 |
> |                                | N = 256                   | 1.00 | 0.94 | 0.72 | 0.39 | 0.23 | 0.16 | 0.14 | 0.18 |
> | One-shot Reasoning w/ Decomp | N = 64                    | 1.00 | 0.96 | 0.97 | 0.98 | 0.94 | 0.91 | 0.86 | 0.83 |
> |                                | N = 128                   | 1.00 | 0.97 | 0.97 | 0.99 | 0.94 | 0.89 | 0.86 | 0.81 |
> |                                | N = 256                   | 1.00 | 0.96 | 0.97 | 0.99 | 0.94 | 0.89 | 0.86 | 0.83 |
>
> **Experiment 2: Reasoning structure alone provides large gains, but self-consistency further closes the gap.** We evaluate agentic reasoning (with and without focused context) at N=1 versus N=64. As shown in Table 2, even at N=1, Agentic Reasoning with Focused Context achieves F1=0.97 at depth 100, already a substantial improvement over one-shot reasoning at N=64 (F1=0.18). Adding self-consistency (N=64) further improves performance to a perfect F1=1.0. This confirms that the performance gain comes from both reasoning structure and higher inference budget.
>
>
>
> Table 2: Agentic reasoning (with and without focused context) on different self-consistency sampling
> | Methods                                | Self-consistency sampling | 2    | 5    | 10   | 20   | 40   | 60   | 80   | 100  |
> |----------------------------------------|---------------------------|------|------|------|------|------|------|------|------|
> | Agentic Reasoning                      | N = 1                     | 1.00 | 0.98 | 0.98 | 0.98 | 0.93 | 0.88 | 0.75 | 0.50 |
> |                                        | N = 64                    | 1.00 | 1.00 | 1.00 | 0.99 | 0.99 | 0.96 | 0.95 | 0.89 |
> | Agentic Reasoning w/ Focused Context | N = 1                     | 1.00 | 0.96 | 0.97 | 0.99 | 0.99 | 0.99 | 0.97 | 0.97 |
> |                                        | N = 64                    | 1.00 | 1.00 | 1.00 | 1.00 | 1.00 | 1.00 | 1.00 | 1.00 |
>
>
> **2. On different sampling strategies across model families**
>
> We acknowledge the difference and clarify our rationale.
>
> (1) First, **our primary analysis focuses on within-model comparisons across reasoning paradigms rather than cross-model rankings.** Under the same model, we use identical sampling settings, ensuring fair comparison.
>
> (2) Proprietary models (GPT-5, Gemini-2.5-Pro) are black-box systems whose internal strategies (e.g., best-of-N, internal self-consistency) are unknown and not configurable. Self-consistency is therefore applied only to open-source models where we have full control over the decoding process. Moreover, in Table 1 above, we also demonstrate that scaling only the self-consistency sampling for one-shot reasoning does not yield a meaningful gain. We will clarify this rationale in the revised manuscript.
>
> ## Q2. Are the reported gains due to improved model reasoning or external scaffolding?
>
> The reported gains should be interpreted primarily as system-level improvements from external scaffolding rather than changes in the base model’s intrinsic reasoning. In particular, the scaffolding helps elicit and stabilize reasoning that is otherwise unreliable by explicating symbolic structure, enforcing intermediate verification, and reducing context interference. Our goal is to decouple these issues: ConvexBench exposes base -model failures under deep composition, while our methods evaluate that structured inference-time scaffolding can mitigate them. We will clarify this distinction in the revised manuscript.

---

> > ### Author Rebuttal · Reviewer_fvkS · 2026-04-04
> >
> > The author explained it very clearly.

---

> > > ### Author Response · Authors · 2026-04-04
> > >
> > > Dear Reviewer fvkS,
> > >
> > > Thank you very much for your positive feedback and for raising the score. We are delighted to hear that our explanations and additional experiments addressed your concerns.
> > >
> > > We will carefully incorporate these discussions into the final manuscript. Thank you again for your constructive review, which has significantly improved the quality of our work.
> > >
> > > Best regards,
> > >
> > > Paper13296 Authors

---

### Decision · Program_Chairs · 2026-04-30

**Decision:**

Accept (regular)

**Comment:**

The paper introduces a benchmark for testing whether LLMs can identify convexity in symbolic expressions and proposes a divide-and-conquer agent that recovers near-perfect performance at extreme composition depths. The reviewers all agree that the problem is well motivated (a clean diagnostic of long-horizon symbolic reasoning), the benchmark is novel and valuable, the exposition is clear and the technical contribution is sound. There were initial concerns about generality, budgeting and the dependence on parsers, but the authors' rebuttal with additional experiments resolved those concerns convincingly. The rebuttal contained matched-budget experiments, sensitivity analyses for parser granularity, detailed cost and latency analyses; the reviewers acknowledged that these additions fully resolved their main concerns.
The reviewers found the failure-mode analysis and the ablations to be convincing, while the agent/method novelty is incremental (composing existing ideas into an effective scaffold).
Overall, the paper describes a rigorous, reproducible benchmark and an agent scaffold that together expose and mitigate a compositional reasoning gap in LLMs.